# Exploring the potential for children to act on antimicrobial resistance in Nepal: Valuable insights from secondary analysis of qualitative data

Jessica Mitchell[1]*, Paul Cooke[2], Abriti Arjyal[3], Sushil Baral[3], Nichola Jones[1], Lidis Garbovan[2], Rebecca King[1]

1 Nuffield Centre for International Health and Development, Faculty of Medicine, University of Leeds, Leeds, United Kingdom, 2 Centre for World Cinema and Digital Cultures, Faculty of Arts, Humanities and Cultures, University of Leeds, Leeds, United Kingdom, 3 HERD International, Kathmandu, Nepal

* j.mitchell1@leeds.ac.uk

## Abstract

This study explores the perceived roles of children in antimicrobial resistance (AMR) in two sites across Nepal. AMR is a global challenge and underpinned by many complex behavioural drivers including how antimicrobial medicines are sourced and used. Because of this social dynamic, several research groups are using community engagement (CE) approaches to understand AMR at community level. However, most data negate the importance of children in behaviours linked to, and potentially driving AMR. In this study, authors apply secondary analysis methods to 10 transcripts representing the views of 23 adults engaged in an AMR-focused film-making project. By focusing on participants' reference to children, we reveal that antimicrobial usage and adherence to health providers' messages can be influenced by the age of the patient. Secondly that children are involved in some of the behaviours which are known to drive antimicrobial resistance such as purchasing over-the-counter antibiotic drugs. Finally, community members discuss that, with careful creation of resources, AMR could be meaningfully presented in educational settings with a view to children acting as agents of change around AMR-driving behaviours. Findings suggest that age-inclusive community engagement projects could be effective in tackling AMR at community level in Nepal and other low resource settings.

## Introduction

Antimicrobial resistance (AMR) refers to the process by which microbes change to survive the drugs used to treat them. This natural process is exacerbated by antimicrobial misuse across human and animal sectors [1]. Taking antimicrobials when they are not needed, such as during a viral infection, or as growth promotors or prophylaxis, places pressure on microbes to change (evolve) and resist the effects of the drug [2, 3]. Factors such as pollution and changing climatic conditions also speed up microbial evolution and can lead to AMR [4–6]. Without

**Data Availability Statement:** All relevant data are within the paper and its Supporting Information files.

**Funding:** Grant Award number AH/T007915/1. Awarded to Jessica Mitchell January 2020 by the AHRC/GCRF Follow-on Funding Call. Original study funded by AH/R005869/1 awarded to Paul Cooke in 2017 by the AHRC. The funders had no role in study design, data collection and analysis, decision to publish, or preparation of the manuscript.

**Competing interests:** The authors have declared that no competing interests exist.

action this decade AMR is predicted to cause 10 million deaths per year by 2050 [7], the bulk of which will occur in low- and middle-income countries [8]. Indeed in 2019 bacterial AMR was shown to be directly attributable to almost 1.3million deaths, and global death tolls were biased toward LMICs [9]. This inequality occurs for financial and systemic reasons, such as limited health care budgets and lack of good sanitation. It is also strongly linked to behavioural actions, namely the availability of antimicrobial medications without prescription, and the lack of awareness of AMR and its consequences [7, 8, 10–12].

Thus, tackling AMR is a social and behavioural challenge [13] which requires engagement with stakeholders at all levels from clinicians to policy makers to the public. The majority of World Health Organisation supported National Action Plans (NAPs), leading global guidance on AMR, stress the need to increase public engagement with the issue of AMR [14]. There are also growing numbers of research projects which work with prescribers, communities and the public to share knowledge on AMR to change AMR-driving behaviours, and create locally meaningful solutions to tackle local barriers to addressing AMR [15–18]. The importance of equitable and meaningful engagement with community members cannot be stressed enough [19], simply raising awareness about the issue of AMR is not enough to change behaviour [16, 20]. Additionally, community-generated data from such projects can be used very effectively to challenge policy and practice at local and National levels, thus creating meaningful long-term change [21–23].

However, even such equitable examples of community engagement can be limited in terms of their reach. Most studies work with adult participants of working-age and those considered to be heads-of-households [18, 24, 25]. Whilst this is important considering that such people are likely to be in decision-making positions in terms of health seeking behaviour and spending on medications, this approach misses the knowledge, attitudes, and practices of younger community members. Recent studies are now beginning to consider young people's knowledge of AMR [26] usually pre- and post- some form of educational intervention. Examples in the UK include an educational theatre play 'The drugs don't work' based on educational resources for 15–18 years old students to engage adult audiences with AMR [27] and an evaluation of school children's knowledge on AMR following their participation in a musical 'The Mould that Changed the World' [28]. Additionally, a growing number of interventions are now geared toward spreading AMR awareness to school age groups (5–18) globally (Table 1).

Two literature reviews suggest that educational interventions targeted at parents and children can be effective in terms of developing Knowledge on AMR [29, 30]. For example, the Roll Back Antimicrobial Resistance (RBA) Initiative demonstrates the potential for youth to act as agents of change acting on AMR awareness in Tanzania through school activities [31]. Alternatively, co-produced AMR animations based on children's drawings appear to have a

**Table 1. An example of the growing number of educational and engagement interventions designed to discuss AMR with children and young people across the globe.**

| Intervention | Scope | Region/Country | Link |
|---|---|---|---|
| Roll-back Antibiotic resistance | Non-governmental Organisation working to tackle AMR via behaviour change including school-based interventions | Tanzania | https://rbainitiative.or.tz/ |
| E-bug | UK Health Security Funded education programme providing school-based resources on microbes, infection prevention and AMR | UK and global translations | https://www.e-bug.eu/ |
| Superheroes Against Superbugs | Public engagement programme using creative approaches to engage young people with AMR and promote AMR action. | India | https://superheroesagainstsuperbugs.com/ |
| Alforja Educativa | Materials for teachers and facilitators to discuss the topics of AMR, microbes and infections with young people | Ecuador | https://alforjaeducativa.reactlat.org/ |
| Youth Against AMR | A collaboration of young people across Kenya, Nepal, Vietnam and Thailand co-creating resources to explain AMR to other young people | Global | https://www.youthagainstamr.com/ |

positive impact on parents' AMR knowledge, attitudes, and beliefs in Ghana [32]. However, there is often limited rationale for these studies, other than 'to tackle the global challenge of AMR'. We understand relatively little about how young people and children are involved in AMR-driving behaviours and AMR action, particularly, in LMICs. This has implications in terms of effective study design, delivery, and evaluation as we may be missing the specific roles and interactions that children and young people have with AMR in their daily lives, plus their potential for AMR action and agency.

This article aims to take a step back from AMR educational interventions and instead consider the specific roles children and young people play in AMR-driving behaviours. This will be achieved by secondary analysis of a community-level dataset collected in Nepal in 2018 (known as The CARAN project) of which we ask: What is the community's perception of young people's potential for acting on AMR in Nepal?

The CARAN (Community Arts Against Antibiotic Resistance Across Nepal) project [33] was a participatory video [34] intervention delivered by the University of Leeds (UK) and HERD International (HERDi), a Nepal-based health research organisation. CARAN took place in 2018 across two sites: Chandragiri municipality represented a peri-urban settlement in Kathmandu and, Lockanthali, a town in Bhaktapur, represented an urban settlement in Nepal. Site selection for this pilot project was purposive; based on proximity and familiarity to the in-country delivery team at HERDi and on initial stakeholder engagement workshops as per the intervention manual [35].

## Methods

This study uses a thematic secondary analysis of transcript data with the aim of conducting a more in-depth analysis of themes from the primary study that researchers considered valuable but were not sufficiently focused on in the primary analysis [36]. Hence, this study uses pre-existing qualitative data to explore a new research question [37]. This approach is similar for instance to other secondary analysis research [38] that explores the roles of men and women in maternal and child nutrition in urban South Africa using a qualitative secondary analysis to investigate new research questions and concepts that emerged from the primary research but could be fully explored only in the secondary analysis of the qualitative data.

The researchers who contributed to this study are aware that secondary data analysis using qualitative data has faced criticism. There are ongoing debates regarding the rigour of and potential methodological and ethical issues arising from such research [36]. The present article addresses these concerns and provides transparent and explicit information about the background and methods of the primary study, the process used to carry out the secondary data analysis and the role of researchers who were part of this project, as well as the ethical considerations involved. Research using secondary analysis of qualitative data stresses the need to outline this process clearly. For instance, Long-Sutehall et al. argue that during the process of carrying out a secondary analysis, the authors must follow several steps: (1) outline the original study, (2) present the process of data collection and (3) highlight the analytical processes applied to the secondary data [39]. In the following section we present the background to the original study and the methodological process of the secondary data analysis, using the guidelines from the literature summarised above.

### Primary study–Background and methods

CARAN used participatory video [34] to create community-level solutions to address the problem of antibiotic resistance in Nepal [33]. A multi-disciplinary team trained 20 adult participants in filmmaking and supported them to develop videos to communicate the challenges

they faced around antibiotic drugs and resistance. The intervention resulted in the co-production of six short films on a range of AMR topics including the misuse of antimicrobials in agriculture, tensions between pharmaceutical and traditional medicine, and non-prescription purchasing of human antibiotics [40]. Participants shared their learning and films in community showcasing events which allowed other members of their family and social circles to gain knowledge on antibiotic use and AMR. Following these intended outcomes, a key discussion point with both participants and wider stakeholders was that the films should be shared with school students to educate them and raise their awareness of AMR. This discussion point inspired the researchers to conduct a secondary data analysis and write this article focused on the potential for children to address AMR challenges in Nepal.

It is important to state that the original project was not designed to capture the roles of young people in AMR. However, secondary analysis of transcript data was conducted to provide insights on how participants felt children could become involved in AMR, which is free from observer and participant bias.

The films are now valued resources used to raise awareness of drug resistance within participants' communities, and to engage policy makers with specific community-level AMR challenges. The films are important also because their content pointed to the significant role that children can play in communities' health seeking behaviour.

A process evaluation was conducted alongside the films with participants invited to focus group discussions (FGDs) and semi-structured interviews (SSIs) during filmmaking and showcasing stages. All activities were conducted in the local language by experienced facilitators based at the HERD International offices in Kathmandu. FGD participation was optional, required an additional consent taking process and did not impact on participant's ability to be involved in the film making project. FGD moderators were experienced personnel from HERDi who had built existing relationships with these communities during the CARAN project. This enabled participants to feel comfortable and speak freely during the FGDs. Audio recordings of workshops, interviews and FGDs were transcribed and translated from Nepali language into English by linguists with fluent Nepali and English.

A full analysis of CARAN data, which includes the films and discusses broader findings and AMR behaviours than those which related to children, can be accessed here [33]. For full details of the primary project methods, including workshop content, observation toolkits, initial stakeholder and participant engagement and consent taking, please see the user manual which can be freely accessed [35].

## Ethical considerations

For the primary study, at both sites written community consent was first acquired from key stakeholders including local stakeholders in the municipality and local government. At the stage of individual participant engagement, poster and word-of-mouth advertisement was used to attract participants who then gave written consent to take part in the film-making process. Ethical approval for this project was granted in 2017 from both the University of Leeds and the Nepal Health Research Council (NHRC) (Ref: 211 2018).

The consent forms used for the primary study (CARAN) specifically discuss the way that transcript data will be used by UoL and HERDi for future analysis of AMR related behaviours. Hence, the consent forms secured consent for future use of data for research purposes, including the secondary analysis that forms the basis of this article. Data confidentiality was ensured by anonymising all the transcripts and removing any personal data that could make the participants identifiable. The anonymised transcripts were saved on password-protected computers that only the research team had access to ensure data protection.

**Table 2. Key terms searched for within the first stage of thematic analysis.**

| KEY TERM | JUSTIFICATION FOR INCLUSION |
|---|---|
| CHILD[1]* | Wildcard term will capture all words relating to stem of child |
| SCHOOL/STUDENT | Captures the position of children within the education system |
| SON/DAUGHTER | Allows gender specific terminology around children to be captured |
| FAMILY | Children may be referred to when mentioning the wider family unit |

[1]Youth and Young* were also included in preliminary searches but these terms were not utilised by participants.

## Qualitative secondary analysis–Methods

For the purpose of this secondary data study, all 10 transcripts from the primary CARAN project were analysed by inductive thematic methods following the guidance of Braun, 2006 [41]. This included six transcripts detailing workshop activities during which original participants created AMR storylines and co-produced their short films. The additional four transcripts capture focus group discussions with these same participants plus local stakeholders and community members who all watched the films at community showcasing events. The views of 23 community members are captured within these transcripts. Film content is discussed throughout the transcripts along with community and stakeholder experiences of AMR. A single researcher (JM) read all transcripts to determine the language used when referring to young people and children. From this initial read-through a set of key search terms was produced (Table 2).

In the second analytical phase, the Microsoft word search function was used to identify anecdotes involving each key term. From now on we will utilise the term children rather than young people, in relation to this dataset as this is the term the participants used. When key terms were identified the entire anecdote was copied into an analytical table to give full context to the way each term was utilized. Following this data extraction step all anecdotes were re-read to determine the themes of discussion involving children.

Researchers (JM and NJ) added notes to the table considering the role or behaviour the child took in each extract. During this step the researchers (JM and NJ) would return to the full transcript if the context of the anecdote needed clarifying, for example referring back to the question the participant was answering when discussing the child/children. Once all data had been fully annotated the researchers (JM, NJ and AA) re-read the table and consolidated the roles of children into broad themes. Finally, this thematic step was repeated to consolidate the roles (Table 3).

**Table 3. The four key roles children take in relation to AMR according to CARAN participants.**

| ROLE | CONTEXT |
|---|---|
| WEAK/ VULNERABLE | Refers to when children are ill. Participants discuss how children require diligent care when ill because they are more *vulnerable* than adults. This includes the help of medical professionals and adherence to prescription and drug usage guidance. |
| PASSIVE | Children may be complicit within behaviours that drive AMR, such as the purchasing of non-prescription antibiotics, but this role is *passive* as they are instructed to do so by family members. |
| ABLE | Children are considered *able* to comprehend AMR education and awareness raising information. Participants suggest the CARAN films and wider AMR information should be delivered in school as part of the curriculum. |
| AUTONOMOUS | Children are considered *autonomous* thinkers with the potential to act as change makers on AMR. Participants believe that after learning about AMR in schools, children have the potential to influence the AMR-behaviour of their household. For example, convincing parents to see a doctor instead of seeking to buy antibiotics without a prescription. |

## Results

Findings suggest the adult participants view children in four complex roles when it comes to AMR-related behaviour. Firstly, children are frequently described as being weak and vulnerable when ill. Interestingly however, this vulnerability appears to enforce adult adherence to medical guidance (including appropriate antibiotic usage via prescription) which parents admit they may not follow if they themselves are unwell.

> *Moderator: There is the practice of using the same medicines for the livestock amongst the general people. They say that my cow got better using this medicine so you can take this.*
>
> *Female participant: They will hand each other the remaining medicines hoping that it will help the other person's cow to get better as well.*
>
> *Moderator: And do they do the same when it comes to children as well?*
>
> *All the participants: No, they do not do so when it comes to children.*
>
> *Moderator: Why do you think that they do this for the adults but not for the children?*
>
> *Female participant: The children are weak.*
>
> \*\*\*
>
> *Female participant: They think that the children are very young and that it might harm them. It might not work on children but it will be useful amongst the adults. They think that the children might have to suffer in case the dose is not correct for them.*
>
> \*\*\*
>
> *Moderator: You think that the children are weaker than the adults while the adults can get better using the same medicines that other adults are using? That is the mentality that the people here have?*
>
> *Female participant: Yes.* (W3)

Children appear as passive recipients of treatment with little autonomy over this process. Nevertheless, it is interesting to note that their perceived vulnerability is enforcing positive AMR behaviours in their parents. Additionally, one of the resulting films exemplifies this point by demonstrating both 'bad' and 'good' scenarios of attending a pharmacy. In the 'bad' scenario an adult male demands non-prescription antibiotics and is angry when his request is refused. Whilst in the 'good' scenario an adult male accompanies his sick adolescent brother to a pharmacy and a diagnostic test is conducted, days later the adult returns to collect the results and medication. This shows, via storytelling in the film, a clear adherence to good health practice (antibiotic stewardship) when a child is concerned. However, the adult male speaks on the child's behalf and leads all aspects of the interaction suggesting the child is both vulnerable and passive.

Transcript data discusses negative AMR-behaviours within the focal communities, including failure to complete full doses, usage without prescription and the above example of sharing antimicrobials between family members and livestock. This behaviour has been perceived as negative during the participatory film-making process (in the original CARAN study) with participants suggesting they would avoid such behaviour because they understand that it can lead to AMR. However, based on the above extract it appears that community members did initially associate such behaviours as being risky prior to the CARAN intervention. Participants discuss how they refrain from sharing antibiotics with sick children because they know

it could cause harm, and the vulnerability of children heightens this risk of harm. Such insights suggest that communities do have working knowledge of negative AMR behaviours, and risk assess where they should and should not action them. Considering children as vulnerable and weak appears to support positive AMR behaviours. However, equally, such perceptions remove any autonomy from the child, as this emphasizes the fact that it is the adults who make the health-related decisions.

Another AMR behaviour discussed in transcript data is the purchasing of antimicrobials from pharmacies without prescription.

*Participant*: *I used to self-determine the antibiotics for the illness that I experienced. And I used to buy them and take them on my own. I even used to instruct my children to buy the medicines for me.* (FGD 5)

Such an example could suggest children are complicit within a negative AMR behaviour. This is also exemplified in one of the co-produced films. Here a male child is given money to purchase antimicrobials from the pharmacy. Interestingly, this is not part of the main storyline but a peripheral interaction suggesting that this type of behaviour is normalized. However, returning to transcript data, the participants describe children being 'told' or 'instructed' by older family members to carry out the purchasing. Indeed, in the film, the young boy is instructed by an adult to go to the pharmacy for antimicrobials. He quickly takes the money and leaves, with no suggestion that he may be able to challenge this instruction. This again removes autonomy from the child and suggest they again take a passive role. In extracts refer-ring to the context in which

children are tasked with purchasing antimicrobials; participants indicate that the medica-tion is for an adult in the household and not the child themselves. This reinforces the previous point that adults do not always adhere to medical guidance around antibiotic use for their own personal health and in the process can confine children to vulnerable and passive roles.

In contrast, other extracts place children in active roles. Several extracts suggest that AMR information such as the CARAN films should be shared with children in school. These discus-sions substantially re-frame the perceived role of children within AMR, with participants con-sidering children *able* to comprehend AMR education and act upon the information they receive.

Participants claim school-based AMR education could raise children's awareness and knowledge of AMR, with the aim of creating antibiotic purchasing and usage changes in their households.

*Local stakeholder*: *If we can specifically take it (CARAN films and programme) to the school sectors then these messages will travel to the community faster. Each household will get this information from there. . ..I think that almost all the households within this municipality will get the message that it gives. That is because every guardian has a child that goes to one of these schools. Plus, there are teachers there as well. The teachers will also explain these things to their students so that they can understand it better. Each household will then get that mes-sage. Along with that, we could also provide them with a small pamphlet each so that they can take home. The people back home will also understand it. Even if we cannot distribute such materials, they will have the experience of watching the documentary and understanding the message that it gives. I think that is a faster way to raise awareness [on ABR].* (SSi 5)

*Moderator: All right. . . who are the people that should be shown those films? Which sector should we also focus on?*

*Participant 2: We should actually focus the children . . . with this programme. It would be better if we could focus on them.* (FGD 4)

Participants agree that if AMR information is tailored to the age of children and taught in schools, children will be able to understand information regarding antibiotic use and the development of AMR. Discussions around AMR education for children tend to appear when moderators ask questions about how the CARAN films could be share more widely. Participants are unanimous in suggesting children as a key new audience, particularly in terms of their ability to influence the adults in their lives. Participants also stress the need for the education to occur in schools where it can be tailored to age.

*Moderator: And students from which grade should be shown these films?*

*Participant 4: . . .I think that students from grade 1 and higher should be shown the film. Most of the families have the same practice of using the medicines again when they fall ill later. They will get some information looking at it and will go home and tell their families about it. They could tell their friends and family back in the village about it which would be good.* (FGD 4)

As mentioned in the above extract, this more active role for children goes beyond recognising their ability to learn about AMR. Children, in this context, are also considered to have power and *autonomy*. These extracts champion the potential of children within positive AMR behaviour because participants view them as change-makers within the community. To exemplify this, one extract from an 18-year-old student attending the film showcasing, links back to conversations regarding unregulated purchasing of antibiotics. This participant suggests children have the potential to challenge such negative AMR behaviours should they be appropriately educated about AMR in school.

*Moderator: And students from which grade should be shown these films?*

*Participant: What should I say. . .? It would be better if everyone was shown this film. However, the young children might not be able to understand it. So, I think that it would be better if we could show it to students that are studying in grade 1 and higher. The children studying in the kindergarten might not understand anything. That is all. (*Note: says shying away)*

(FGD 4)

*Moderator: So you said that your family members send you to get their medicines. At what age do they feel that you are able to understand what medicines they are referring to? I think that it will be easier to determine the age in this manner.*

*Participant: When we are 10 years old.*

*Moderator: So it would be better to show it to the children of ages 10 and above? (referring to school AMR education)*

*Participant: Yes. They will be able to understand it.* (FGD 4)

## Discussion

At both the urban and peri-urban regions sites, participants consider children to be more vulnerable to illness than adults. As such there appears to be potential to leverage this perceived

vulnerability to encourage prescription and medical practitioner visits prior to antimicrobial use. The demand for, and use of non-prescription antimicrobial medicines is high across Nepal [11, 12, 42, 43]. This is part due to complexities in identifying and costs of visiting qualified health care providers [1, 11]. Many community people prefer to directly purchase antimicrobials from pharmacies which are often staffed by unqualified practitioners and/or are under pressure to make financial profits [11, 43].

As mentioned, one of the films shows a clear adherence to good health practice (antibiotic stewardship) when a child is concerned but disregards it when an adult is seeking antibiotics for themselves. Vulnerability is recognized to influence compliance with other health issues [44] for example, smoking cessation interventions worldwide often focus on the risks smoking poses to young children [45–47]. This extends to climate-related health issues across sub-Saharan Africa where the use of clean cooking stoves is being targeted toward women with children because the air pollution from traditional kerosene stoves predominantly harms the development of foetuses, babies and young children [48] as well as the environment. Tapping into vulnerabilities is thus often perceived as a useful tactic in the promotion of safer behaviours for humans and the environment, something which could potentially be expanded to AMR.

Alongside being vulnerable, transcript data initially places children in passive roles, as the recipients of treatment sourced by adults. Children's passivity is reinforced through the co-produced films; for example, an adult male is shown leading all dialogue around the safe treatment of his sick adolescent brother. The film suggests that it is normal for a child to have an adult family member speak on their behalf, rather than being active within their own healthcare.

In another film a child is handed money and directed to the pharmacy to purchase medicines. Transcript data also discuss the role of children in purchasing non-prescription antimicrobials under the instruction of adults. This behaviour is particularly concerning as it can directly contribute to AMR. If antimicrobials are sourced without medical consultation and/or diagnostic testing it is unlikely that the most appropriate treatment will be obtained [49–51]. Using an unsuitable antimicrobial means the presenting illness will not be treated and could become more serious. Other microbes in the body will also be challenged by the antibiotic and could evolve resistance which in turn will make them less responsive to treatment next time they cause an infection [52].

Data which describes children purchasing non-prescription antibiotics cannot be generalized across Nepal or to other LMICs, but they are important. They provide evidence that children contribute to behavioural drivers of AMR, an issue which may warrant further investigation at National and Global level. Other sources show that non-prescription purchasing is common practice amongst adults in Nepal and across South-East Asia [43, 53–56]. But the role of children in this behaviour is poorly understood [57]. In our dataset, children appear to be passively following the orders of their parents, which is unsurprising considering the hierarchical society of Nepal where it is unlikely that children would challenge the instruction of their elders [58]. Understanding this context suggests that any AMR education offering would have to focus on developing children's confidence and autonomy to safely refuse requests from adults. Nevertheless, learning that children do have a role in non-prescription purchasing provides a meaningful and relatable experience on which to target future AMR interventions.

## Challenges and potential for children to act on AMR

Engaging children directly may be the most impactful approach to minimising non-prescription antimicrobial purchasing. This is because children are already engaged in very rapid

learning through their education [59]. They have fewer pre-conceptions of what is 'normal' than do adults and are thus potentially better able to challenge the status quo and imagine new approaches to tackling problems. This understanding has facilitated many successful educational initiatives aimed at tackling unsustainable environmental interactions which contribute to the climate crisis [59, 60]. Thus, despite being perceived as passive within AMR, Nepali children do have behavioural experience to relate to, and this could be explored through specific engagement interventions which tap into children's' rapid learning potential.

Indeed, the most frequent comment from the adult participants is that AMR could be addressed by the education of children. Transcripts show that participants repeatedly describe children as being able to learn about AMR if age-appropriate material is delivered in schools. As transcript discussions on education evolve, multiple participants suggest that AMR-aware children will autonomously share their learnings at home and influence positive AMR behavioural changes in their families and wider communities. This view has clear parallels with climate-crisis education, where a growing body of evidence now suggests that climate-aware children both take autonomous action and encourage behavioural changes at societal level [60–62]. This is particularly so in the literature relating to climate-induced disaster education: when children are taught about risk, they can share this information with their family and wider communities to better support mitigation strategies [61, 63, 64]. In other settings such as WaSH (water, sanitation, and hygiene) there is a generic trend to discuss children as 'agents of change' in terms of promoting good sanitation practices. However evidence is patchy and studies finding that such messages can be effectively conveyed to household members reflect on the need for interventions to be context specific [65].

Indeed, there are several obstacles in terms of Nepali children becoming change-makers on AMR. Firstly, the data reveals a complex set of interrelated issues and structures which underpin children's roles in AMR. Although children are considered able and autonomous to act on AMR, they also hold passive or submissive roles in society which raises questions over how realistic it is to expect children to challenge the instructions of their elders. Nepali society is very respectful of older generations and age-based hierarchies, thus even with AMR education and awareness children may not have the agency to refuse this type of instruction [59]. It is difficult to see how education could create wider behavioural change on AMR if children have no agency to act on the information they have learned. However, Corcoran suggests that children can be active in 'creating their own cultures and life world' if they are appropriately supported to develop their own opinions and attitudes toward environmental issues [66]. Article 12 of the UN declaration on the rights of the child echoes this sentiment by emphasising that children must have the right to shape the services that support them. In practice this will often mean educating children on a specific issue (such as AMR) but also providing support to develop critical thinking patterns and confidence to suggest changes.

Secondly, and linked to the first obstacle, AMR interventions which just focus on awareness raising have failed to enact meaningful behavioural change in their focal communities [16]. This echo's the view of Corcoran, 2009 [66] that children can become agents of change within challenging dialogues but only if they are encouraged to develop their own attitudes toward these issues through social processes including family/community dialogues and pedagogical approaches. This echo's the definition of Community Engagement co-created by members of the behind the original CARAN project.

*A participatory process through which equitable partnerships are developed with community stakeholders, who are enabled to identify, develop and implement community-led sustainable solutions using existing or available resources to issues that are of concern to them and to the wider global community.* [18].

Applying this approach to AMR education could ensure children not only learn but feel empowered and confident to share this learning and thus facilitate wider behavioural change in their community. AMR is a problem many children in Nepal will already experience, through their purchasing of non-prescription antimicrobials as previously discussed. Our films and transcripts also suggest that children witness antimicrobial (mis)use in small-holder farming. Thus, with careful explanation these relatable behaviours could be explored within AMR education interventions. The use of community-produced resources would also be helpful in terms of localising and relating the problem to children. The communities we have been working with are keen for the CARAN films to be used as teaching resources, while the participants have also expressed an interest in supporting educational offerings. This is likely to be an effective approach because a growing body of evidence suggests that participatory and creative methods of learning are impactful and actually lead to attitudinal and behavioural change more often than knowledge gained in isolation [62, 67–69].

## Limitations

The secondary data used in this thematic analysis is small and represents just two discrete communities, but it is interesting to realize how participants weigh up the risks and benefits of engaging in AMR-negative behaviours. Understanding this process in more detail represents a useful route to create AMR behavioural change. For example, health professionals could frame AMR guidance to better highlight human vulnerabilities to drug resistant infections at all ages. Or, another example, pharmaceutical practices could be discouraged from serving individuals under a specific age, for example by a financial penalty. Nepal, like many countries, has a National Action Plan on AMR and regulating the sale of pharmaceuticals is discussed within this document. Of course, developing and enforcing guidance are distinct strategies, and it may be very difficult to identify who is old enough to purchase medications or to enforce any form of ID checking or penalty fines in situ.

## Directions for future research

The engagement of young people on AMR represents a fruitful opportunity for future research at local and national levels in Nepal, but also globally as the world's youth face similar AMR challenge. Indeed, in late 2020 an open letter calling for the World Health Organisation (WHO) to ring-fence funding to establish a "United digital online platform to support engagement of young people on AMR" was signed on behalf of 22 organisations across Africa, Asia, and the UK [70]. Engaging young people with the issue of AMR via community engagement is certainly possible at grass roots level as evidenced by many groups who signed this. This was presented for discussion at the 148th session of the WHO's executive board in January 2021 [71]. However, the only recorded action relating to AMR and young people is the acknowledgement that world antimicrobial awareness week (WAAW) is timed to fall over November 21st which is also World Children's Day. Moreover, educational materials must be contextually appropriate for the age group and setting in which they are to be delivered and at present our wider understanding of young people's role in AMR is limited. As such, we suggest that more detailed engagement with young people directly, and their communities, is needed to understand their roles and behaviours in relation to local AMR challenges. Such information can then support the co-creation of appropriate AMR educational and engagement materials for use with different age groups in school or extracurricular settings.

In future research it would be interesting to re-engage original participants and younger members of their community to discuss practical ways to engage children on AMR. It would be informative to hear both adult and younger people's perspectives on how children can be

empowered as change-makers despite their ostensibly passive position in family and community structure in this setting. It would be valuable to better understand parents' perceptions of children as vulnerable and how this influences adherence to medical guidance. Such information could help develop more effective public health messaging. Finally, it would be useful to better understand the role of children in non-prescription purchasing of antimicrobials. Is this a passive process or do children question their instructions? How does this process differ based on socio-economical, cultural, and other parameters? Posing such questions could allow young people and their wider community to appreciate the challenges of this behaviour and consider how appropriate it is.

## Conclusions

A secondary analysis of transcripts from the Community Arts Against Antimicrobial Resistance (CARAN) project reveals multiple routes for young people and children to influence the global challenge of AMR. In this setting children were involved in non-prescription purchasing of antimicrobials and could also influence their parents' antimicrobial stewardship decisions. Additionally, there is demonstrable motivation at community level to share the project's resources with school-age children which would be an equitable and sustainable use of existing outputs. However, there is a need to understand in more detail the role of young people in AMR associated behaviours to create educational interventions that effectively promote behavioural change. We suggest that efforts to tackle AMR in Nepal, and other LMICs, would benefit from the direct engagement of young people. Understanding their knowledge, attitudes, and practices regarding antimicrobial use, AMR driving behaviours and AMR agency is essential to shaping AMR educational interventions and increasing their impact going forward.

## Supporting information

**S1 File. Transcript inventory.**
(DOCX)

## Acknowledgments

Authors would like to thank the communities who took part in the initial research project on which this analysis is based. We particularly extend our thanks to the Ministry of Health and Population, Nepal who supported the original study and attended stakeholder workshops during and after the initial research was conducted. We also thank the additional researchers at HERD International and the University of Leeds who assisted with data collection and project delivery. We extend a special thank you to Ashim Shreshta for his film-making expertise during the research project in Nepal.

## Author Contributions

**Conceptualization:** Jessica Mitchell, Paul Cooke, Sushil Baral.

**Data curation:** Jessica Mitchell, Abriti Arjyal.

**Formal analysis:** Jessica Mitchell, Abriti Arjyal, Nichola Jones.

**Funding acquisition:** Jessica Mitchell, Paul Cooke, Sushil Baral, Rebecca King.

**Methodology:** Jessica Mitchell, Paul Cooke, Abriti Arjyal, Sushil Baral, Rebecca King.

**Project administration:** Jessica Mitchell, Abriti Arjyal.

**Supervision:** Paul Cooke, Sushil Baral, Rebecca King.

**Writing – original draft:** Jessica Mitchell, Lidis Garbovan.

**Writing – review & editing:** Jessica Mitchell, Paul Cooke, Nichola Jones, Lidis Garbovan.

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
