## [Decision Letter · Decision Letter 0]

15 Mar 2023

PONE-D-23-04751Exploring the potential for children to act on antimicrobial resistance in Nepal: valuable insights from secondary analysis of qualitative dataPLOS ONE

Dear Dr. Jessica Mitchell,

Thank you for submitting your manuscript to PLOS ONE. After careful consideration, we feel that it has merit but does not fully meet PLOS ONE’s publication criteria as it currently stands. Therefore, we invite you to submit a revised version of the manuscript that addresses the points raised during the review process. Please ensure that your decision is justified on PLOS ONE’s publication criteria and not, for example, on novelty or perceived impact.I would suggest to review the spelling errors and response all the comments from reviewers.

We look forward to receiving your revised manuscript.

Kind regards,

Kshitij Karki, MPH, MA

Academic Editor

PLOS ONE

Journal Requirements:

Additional Editor Comments (if provided):

Thank you for this piece of work. Please address the suggestions and comments from reviewers.

I would like to request you to address the spelling errors such as Nepal Health Research Council, name of the places (Lokanthali, Bhaktapur) and so on, along with reviewers comments.

Reviewers' comments:

Reviewer's Responses to Questions

**Comments to the Author**

1. Is the manuscript technically sound, and do the data support the conclusions?

Reviewer #1: Yes

Reviewer #2: Yes

2. Has the statistical analysis been performed appropriately and rigorously? 

Reviewer #1: N/A

Reviewer #2: N/A

3. Have the authors made all data underlying the findings in their manuscript fully available?

Reviewer #1: Yes

Reviewer #2: Yes

4. Is the manuscript presented in an intelligible fashion and written in standard English?

Reviewer #1: Yes

Reviewer #2: Yes

5. Review Comments to the Author

Reviewer #1: The authors have addressed the potential ethical issues of this paper well and I am satisfied that there is no ethical issue. This is a well-written paper on an interesting subject. I would query whether a child would be able to question an adult's behaviour to the extent of refusing to visit a pharmacy to buy OTC antibiotics and although this is mentioned in the Discussion I think it could be expanded on. I do agree that educating children and making them aware of AMR is important, and that their knowledge can be filtered up to their parents and other family members. I think the Discussion should include more context - it is very common to visit a pharmacy in Nepal and buy OTC medications, including antibiotics. Please also discuss the potential for participants to have told interviewers what they thought they wanted to hear (which was an issue in my research in Nepal, particularly in rural areas with less well-educated participants). A broader discussion of the fact that many papers suggest that awareness and engagement doesn't actually work to effect behaviour change would also be useful in the relevant section. Overall an interesting, potentially useful paper.

Reviewer #2: The manuscript describes utilization of qualitative data collected to aid development of an intervention for improved antibiotic stewardship, during which the authors discovered a theme worthy of further analysis. It is great to see qualitative data exploited fully in this way and this methodology is worth highlighting. Some of the data derived on the role of children in antibiotic procurement, in an environment where antibiotics are available over the counter is not new, however, the methodology and demonstration of its value is a contribution to research on this topic and others involving qualitative data collection. At times I had to go back and forth to determine how the data were derived and who was involved in the study, so many of my comments are related to clarity and question the order that the text is presented.

Abstract

While this is a qualitative study, some numbers would be useful to include. Exactly how many participant transcripts were analysed, how many participants were enrolled?

Line 10: Here it is claimed that ‘we reveal that antimicrobial usage and adherence to

11 health providers’ messages are influenced by the age of the patient.’ Reading the text I didn’t find this. Perhaps I missed the adherence data or maybe this is an English expression that doesn’t align with the findings.

Aims

This seems repetitive and long, especially coming after the objective paragraph in the introduction and prior to the methods. I would focus on a brief summary to introduce the secondary analysis overview in the methods. The way it is written is appropriate for a PhD thesis but unnecessary for a research paper. You could then take out any redundancy in the methods e.g. the first paragraph or just remove the aim section and keep the first paragraph of the methods as is.

Methods

Lines 126-132 would fit better in the introduction.

Line 145: the authors describe ‘intended outcomes'. I'm not clear whether drawing out discussion on children's roles was a primary objective of the study or a theme that evolved. Reading ahead it is stated that it was not a primary objective. Review the order of the text to ensure that readers are clear on secondary data analysis versus reanalysing data with a further theme in mind. It does not affect validity of the analysis but is needed for clarity of the methods employed.

Line 153: the authors mention interventions. I'm not clear on what is meant here. is this more about the film making process or were there interventions developed and implemented?

Line 181: Exactly how many participants were included in the primary study. Even though this is not a quantitative paper, it is useful to know without having to download the papers on the primary project.

Line 189: here is the information I was asking about in terms of whether young people's roles evolved as a theme during the study. Move this earlier for improved clarity.

Line 193-207 can be omitted. This is not needed for a research manuscript or place some of the text in the authorship information section.

Table 2: move the second sentence of the title to a footnote.

Results

Line 258: positive behaviour was not suggested by the quote above, where adults admitted that they did not seek prescriptions. Just attending a pharmacy is presumably not a behaviour that would comprise antibiotic stewardship. Presenting to a qualified medical practitioner should be the ideal behaviour.

Line 262: I’m surprised to hear that in Nepal, diagnostic tests are recommended before providing drugs. This seems at odds with elsewhere in Asia and in Africa. Perhaps this is an English expression issue and not what the authors are trying to describe.

Line 267: I cannot tell whether these are results from the previous work or from this analysis focusing on children/young people. I recommend better clarity. Consider including this information should be in the Discussion.

Line 330: to what end would information from the films be useful for children who do not have the agency to decide about antibiotic purchase? To influence their parents?

Line 348: more information on the participant is needed. Is this a parent or a stakeholder? I'm assuming that stakeholders were not from the education sector, as this theme emerged from discussions and was not anticipated.

Line 356: provide more information on the participant. Was this a child?

Discussion

Line 363: the data presented do not seem to support the assertion that child vulnerability promotes adults to seek prescriptions and medical practitioner advice. I think that the authors are trying to say that there's potential to leverage perceived vulnerability to encourage prescription and medical practitioner visits.

Line 365: I did not read about these data in the results. Do not include new data in the Discussion.

Line 379: I didn't see these data in the results. Include film related data in a table and describe in the results.

Line 429: there are examples of children from LMICs being educated on water, sanitation and hygiene practices and whether the messages have been conveyed to household members (one example is Winter JC, Darmstadt GL, Lee SJ, Davis J.BMC Public Health. 2021 Oct 8;21(1):1812).

6. PLOS authors have the option to publish the peer review history of their article (what does this mean?). If published, this will include your full peer review and any attached files.

Reviewer #1: No

Reviewer #2: No

---

## [Author Response · Author response to Decision Letter 0]

18 Apr 2023

Dear editor

We greatly appreciate the comments of two anonymous reviewers on our manuscript entitled Exploring the potential for children to act on antimicrobial resistance in Nepal: valuable insights from secondary analysis of qualitative data. 

This manuscript presents an original qualitative research study to consider the perceptions of adults on the role young people/children can play in AMR driving behaviors in Nepal. We have applied rigorous secondary data analysis methods to a dataset of 10 transcripts from Focus Group Discussions as part of a Participatory video study conducted between 2017-2019. All ethical approvals for this study were granted from both UK and Nepali authorities and the protection of participants was of paramount concern during the original study and secondary analysis. We can confirm that the results of this analysis have not been presented elsewhere although there are multiple publications forthcoming on the original participatory video study, those already published are cited throughout. 

Our specific responses to each comment are presented in the list below. However, we would like to explain in some detail the responses to Reviewer 2’s comments around the discussion of film findings. As explained in our methods section, the films themselves were not analyzed in this study. This is for two reasons; first, analysis of participatory video is a very specific method and is being conducted for other work as a part of this project. Secondly, the transcripts with video-making participants refer to specific storylines in the film which guide us to those extracts and allow us to discuss them in line with the transcript quotes. We feel confident that we have clearly explained when and where this occurs and that the film information makes a valuable contribution to the paper. We hope this is acceptable to yourselves and the reviewer. 

All other comments have been addressed as per the line-by-line details given below and we feel they have contributed to making the manuscript clearer and more engaging to read. All line numbers refer to the file named Manuscript which is the revised version without track changes.

With very best wishes

Dr Jessica Mitchell on behalf of the Authorship Team

Editor Comments 

I would like to request you to address the spelling errors such as Nepal Health Research Council, name of the places (Lokanthali, Bhaktapur) and so on, along with reviewers comments. All amended based on follow-up email with editorial team

Reviewer #1 comments

The authors have addressed the potential ethical issues of this paper well and I am satisfied that there is no ethical issue. This is a well-written paper on an interesting subject. 

I would query whether a child would be able to question an adult's behaviour to the extent of refusing to visit a pharmacy to buy OTC antibiotics and although this is mentioned in the 

Discussion I think it could be expanded on. We have utilised your comments alongside those of the other reviewer to expand the discussion in some areas. However, we would prefer to keep the paper to a reasonable length and feel our current comments on this matter are detailed (lines 424-444).

I do agree that educating children and making them aware of AMR is important, and that their knowledge can be filtered up to their parents and other family members. I think the Discussion should include more context - it is very common to visit a pharmacy in Nepal and buy OTC medications, including antibiotics. Thank you for this comment, we realise this contextual information is lacking in the discussion section and have amended as suggested, see lines 353 – 358.

Please also discuss the potential for participants to have told interviewers what they thought they wanted to hear (which was an issue in my research in Nepal, particularly in rural areas with less well-educated participants). Absolutely a valid point, amendments made to lines 162 - 165. Our FGDs were facilitated by HERDi colleagues who are very experienced but also very familiar to this community which allowed participants to speak freely, ask questions and take breaks. There are indeed quite a few jokes and tangents recorded in the FGDs!

A broader discussion of the fact that many papers suggest that awareness and engagement doesn't actually work to effect behaviour change would also be useful in the relevant section. This is covered in the introduction lines 64-66

Overall an interesting, potentially useful paper.

Reviewer #2: 

The manuscript describes utilization of qualitative data collected to aid development of an intervention for improved antibiotic stewardship, during which the authors discovered a theme worthy of further analysis. It is great to see qualitative data exploited fully in this way and this methodology is worth highlighting. Some of the data derived on the role of children in antibiotic procurement, in an environment where antibiotics are available over the counter is not new, however, the methodology and demonstration of its value is a contribution to research on this topic and others involving qualitative data collection. At times I had to go back and forth to determine how the data were derived and who was involved in the study, so many of my comments are related to clarity and question the order that the text is presented.

Abstract

While this is a qualitative study, some numbers would be useful to include. Exactly how many participant transcripts were analysed, how many participants were enrolled?

we have added the number of transcripts (10) and participants (23) to line 31 of the abstract.

Line 10: Here it is claimed that ‘we reveal that antimicrobial usage and adherence to

11 health providers’ messages are influenced by the age of the patient.’ Reading the text I didn’t find this. Perhaps I missed the adherence data or maybe this is an English expression that doesn’t align with the findings. –We respectfully decline to change the line about antimicrobial usage and age because a key finding of the paper that adults adhere to antimicrobial usage and usage guidance more closely for their children thus based on our analysis we feel confident to say that usage and adherence to health providers messages are influenced by age.

Aims

This seems repetitive and long, especially coming after the objective paragraph in the introduction and prior to the methods. I would focus on a brief summary to introduce the secondary analysis overview in the methods. The way it is written is appropriate for a PhD thesis but unnecessary for a research paper. You could then take out any redundancy in the methods e.g. the first paragraph or just remove the aim section and keep the first paragraph of the methods as is.- Section removed and shortened as suggested, see lines 96-100. 

Methods

Lines 126-132 would fit better in the introduction. – moved as suggested, now lines 101-108

Line 145: the authors describe ‘intended outcomes'. I'm not clear whether drawing out discussion on children's roles was a primary objective of the study or a theme that evolved. Reading ahead it is stated that it was not a primary objective. Review the order of the text to ensure that readers are clear on secondary data analysis versus reanalysing data with a further theme in mind. It does not affect validity of the analysis but is needed for clarity of the methods employed.- order of the text revised – see lines 141-152

Line 153: the authors mention interventions. I'm not clear on what is meant here. is this more about the film making process or were there interventions developed and implemented? – replaced with ‘activities’, now line 191.

Line 181: Exactly how many participants were included in the primary study. Even though this is not a quantitative paper, it is useful to know without having to download the papers on the primary project. – 20 participants were trained within the original PV project (line 134) and the transcripts represent the views of 23 community members including some of these participants but also community stakeholders such as local mayors and community members who attended the showcasing events (lines 32 and 195).

Line 189: here is the information I was asking about in terms of whether young people's roles evolved as a theme during the study. Move this earlier for improved clarity. –Position changed– see lines 141-152

Line 193-207 can be omitted. This is not needed for a research manuscript or place some of the text in the authorship information section. Removed 

Table 2: move the second sentence of the title to a footnote.- Footnote added to page 9

Line 258: positive behaviour was not suggested by the quote above, where adults admitted that they did not seek prescriptions. Just attending a pharmacy is presumably not a behaviour that would comprise antibiotic stewardship. Presenting to a qualified medical practitioner should be the ideal behaviour. This sentence does not refer to the quote but the film example. The sentence in line 236 begins with ‘additionally’ to clearly differentiate between the sources of evidence used. Line 243-253 clarifies that this example comes from film rather than the transcript.

Line 262: I’m surprised to hear that in Nepal, diagnostic tests are recommended before providing drugs. This seems at odds with elsewhere in Asia and in Africa. Perhaps this is an English expression issue and not what the authors are trying to describe. This refers to the scenario depicted in the film created by community members – we recognise that this behaviour may be rare in Nepal. However, it demonstrates the participants understanding of AMR following their participation in the AMR project from which this study was based. Line 236 clarifies that this example comes from film rather than the transcript.

Line 267: I cannot tell whether these are results from the previous work or from this analysis focusing on children/young people. I recommend better clarity. Consider including this information in the Discussion. Finding from original CARAN study now clearly identified as such in line 236.

Line 330: to what end would information from the films be useful for children who do not have the agency to decide about antibiotic purchase? To influence their parents? Clarified in line 317-318. Participants feel children can influence adults in their lives and that by seeing the AMR films this influence could extended to AMR behaviours.

Line 348: more information on the participant is needed. Is this a parent or a stakeholder? I'm assuming that stakeholders were not from the education sector, as this theme emerged from discussions and was not anticipated. Line 331 now gives more information on this Participant, a recent high school graduate aged 18.

Line 356: provide more information on the participant. Was this a child? Line 331 now gives more information on this Participant, a recent high school graduate aged 18.

Discussion

Line 363: the data presented do not seem to support the assertion that child vulnerability promotes adults to seek prescriptions and medical practitioner advice. I think that the authors are trying to say that there's potential to leverage perceived vulnerability to encourage prescription and medical practitioner visits. Absolutely! Thank you for suggesting the wording change this is so much clearer now! Opening sentence to discussion is amended. 

Line 365: I did not read about these data in the results. Do not include new data in the Discussion. We are confident that the film storyline presented in the results section line 306 discusses this in detail (now starting from line 351).

Line 379: I didn't see these data in the results. Include film related data in a table and describe in the results. We have included links to the films but as the content is discussed throughout the transcripts, we did not analyse the films frame-by-frame but were guided to specific sections by the participants discussions in the FGDs. This links back to the comments made and response given around lines 243-253 above.

Line 429: there are examples of children from LMICs being educated on water, sanitation and hygiene practices and whether the messages have been conveyed to household members (one example is Winter JC, Darmstadt GL, Lee SJ, Davis J.BMC Public Health. 2021 Oct 8;21(1):1812). Thank you we have included this reference in the re-framed section lines 422-426.

---

## [Decision Letter · Decision Letter 1]

4 May 2023

Exploring the potential for children to act on antimicrobial resistance in Nepal: valuable insights from secondary analysis of qualitative data

PONE-D-23-04751R1

Dear Jessica Mitchell,

We’re pleased to inform you that your manuscript has been judged scientifically suitable for publication and will be formally accepted for publication once it meets all outstanding technical requirements.

Kind regards,

Kshitij Karki, MPH, MA 

Academic Editor 

PLOS ONE

Additional Editor Comments (optional):

Reviewers' comments:

Reviewer's Responses to Questions

**Comments to the Author**

1. If the authors have adequately addressed your comments raised in a previous round of review and you feel that this manuscript is now acceptable for publication, you may indicate that here to bypass the “Comments to the Author” section, enter your conflict of interest statement in the “Confidential to Editor” section, and submit your "Accept" recommendation.

Reviewer #1: All comments have been addressed

Reviewer #2: All comments have been addressed

2. Is the manuscript technically sound, and do the data support the conclusions?

Reviewer #1: Yes

Reviewer #2: Yes

3. Has the statistical analysis been performed appropriately and rigorously? 

Reviewer #1: N/A

Reviewer #2: N/A

4. Have the authors made all data underlying the findings in their manuscript fully available?

Reviewer #1: Yes

Reviewer #2: Yes

5. Is the manuscript presented in an intelligible fashion and written in standard English?

Reviewer #1: Yes

Reviewer #2: Yes

6. Review Comments to the Author

Reviewer #1: (No Response)

Reviewer #2: The authors have addressed all of my comments and suggestions.

There are some typos among the newly inserted text. I urge the authors to go through and fix these now rather than during the proof process.

7. PLOS authors have the option to publish the peer review history of their article (what does this mean?). If published, this will include your full peer review and any attached files.

Reviewer #1: No

Reviewer #2: No

---

## [Editor Report · Acceptance letter]

24 May 2023

PONE-D-23-04751R1 

Exploring the potential for children to act on antimicrobial resistance in Nepal: valuable insights from secondary analysis of qualitative data. 

Dear Dr. Mitchell:

I'm pleased to inform you that your manuscript has been deemed suitable for publication in PLOS ONE. Congratulations! Your manuscript is now with our production department. 

Kind regards, 

on behalf of

Dr. Kshitij Karki 

Academic Editor

PLOS ONE